# Association between Nutrients and Visceral Fat in Healthy Japanese Adults: A 2-Year Longitudinal Study Brief Title: Micronutrients Associated with Visceral Fat Accumulation

**DOI:** 10.3390/nu11112698

**Published:** 2019-11-07

**Authors:** Naoki Ozato, Shinichiro Saito, Tohru Yamaguchi, Mitsuhiro Katashima, Itoyo Tokuda, Kaori Sawada, Yoshihisa Katsuragi, Seiya Imoto, Kazushige Ihara, Shigeyuki Nakaji

**Affiliations:** 1Department of Active Life Promotion Sciences, Graduate School of Medicine, Hirosaki University, Hirosaki 0368562, Japan; katashima.mitsuhiro@kao.com (M.K.); katsuragi.yoshihisa@kao.com (Y.K.); 2Health Care Food Research Laboratories, Kao Corporation, Tokyo 1318501, Japan; yamaguchi.tohru@kao.com; 3Biological Science Research Laboratories, Kao Corporation, Tokyo 1318501, Japan; saito.shinichiro@kao.com; 4Department of Social Medicine, Graduate School of Medicine, Hirosaki University, Hirosaki 0368562, Japan; i-tokuda@hirosaki-u.ac.jp (I.T.); iwane@hirosaki-u.ac.jp (K.S.); ihara@hirosaki-u.ac.jp (K.I.); nakaji@hirosaki-u.ac.jp (S.N.); 5Health Intelligence Center, Institute of Medical Science, University of Tokyo, Tokyo 1088639, Japan; imoto@ims.u-tokyo.ac.jp

**Keywords:** micronutrients, visceral fat, obesity, BMI, macronutrients, vegetable

## Abstract

High visceral fat area (VFA) is a stronger predictor of cardiovascular disease and overall mortality than body mass index or waist circumference. VFA may be decreased by proper dietary habits. Although previous epidemiologic studies demonstrated an association between nutritional components or foodstuffs and VFA, only the associations of a few nutrients, such as dietary fiber and calcium, are reported. We performed a comprehensive 2-year longitudinal study in more than 624 healthy people and analyzed 33 micronutrients to investigate nutrients that contribute to changes in visceral fat. Our analyses revealed that “macronutrients” and “micronutrients” were “mutual confounders”. Therefore, when evaluating the association between VFA and micronutrients, associations were adjusted by macronutrients. The ingestion of 7 nutrients: soluble dietary fiber, manganese, potassium, magnesium, vitamin K, folic acid, and pantothenic acid, which are abundant components in vegetable diets, was significantly inversely correlated with a change in VFA. Additionally, a change in the ingestion of one nutrient, monounsaturated fat, was significantly positively correlated with a change in VFA. These associations were independent of body mass index and waist circumference. Thus, a predominantly vegetable diet may decrease VFA. In addition, adjusting the intake of macronutrients might help to clarify the association of micronutrients with VFA.

## 1. Introduction

Obesity results from an imbalance between energy consumption and expenditure. Therefore, increases in physical activity or proper dietary habits are recommended for maintaining an appropriate body weight [1]. Dietary habits are crucial in the development of overweight and obesity [2], and suboptimal diet can potentially be a major contributor to mortality in all countries worldwide [3]. Visceral fat accumulation, also known as visceral obesity, is a major predictor of cardiovascular disease [4] and all-cause mortality [5,6,7] independent of the body mass index (BMI) and general obesity. Clustering of metabolic risk factors, including hypertension, high blood glucose and triglyceride (TG) concentrations, and low serum high-density lipoprotein (HDL) cholesterol, is more strongly associated with the visceral fat area (VFA) than high subcutaneous fat, waist circumference (WC), or BMI [8,9]. Thus, evaluation of diets to reduce VFA is important to protect against poor metabolic status.

Some epidemiologic studies, including cross-sectional studies [10,11,12,13,14,15,16,17,18,19] and longitudinal studies [20,21,22,23,24], have investigated the association between nutritional components or foodstuff and VFA. These analyses indicated that only dietary fiber [17,21,23], carotenoid [15] and calcium [16,22] were negatively associated with VFA in terms of nutritional components. There are 2 analyses including over 1000 individuals; however, both studies targeted exclusively Westerners and patients [20,21]. We reported the association of visceral fat and nutrition in healthy Asians who eat mainly rice and tend to accumulate more visceral fat compared with Westerners [25] in a cross-sectional study [12,19].

In the present study, we performed a 2-year longitudinal study in 624 relatively healthy Japanese subjects, and comprehensively investigated the nutrients associated with the accumulation of visceral fat by assessing the relationship of 33 micronutrients and VFA.

## 2. Materials and Methods

### 2.1. Study Subjects

The Iwaki Health Promotion Project was launched in 2005 as an annual health check-up for local residents, aiming to prolong a healthy lifespan. Participants were men and women of at least 20 years of age living in the Iwaki region of Hirosaki City, Aomori Prefecture, Japan [26,27,28,29]. Inclusion criterion was individuals who have health condition ability to participate in this health check-up for more than 5 h. VFA was first introduced as a health check-up parameter in 2015, and the present analyses were performed using data obtained from the health check-ups held from May 2015 to May 2017. In 2015, 1118 individuals participated in the health check-up (Figure 1). Of these, 36 participants did not complete the clinical assessments, dietary data, and/or VFA, and were excluded from the analyses. In addition, we excluded 458 participants with missing dietary and/or VFA data from 2016 and/or 2017. Thus, 624 individuals (260 men and 364 women; mean age ± standard deviation, 54.0 ± 14.2 years for men and 55.1 ± 13.6 years for women) were enrolled into the analyses.

### 2.2. Dietary Exposure

A validated, self-administered, brief diet history questionnaire (BDHQ) was used to assess the dietary intake of the participants during the preceding month [30,31,32]. The BDHQ is a structured questionnaire that contains questions about the intake of approximately 58 foods and beverages, and allows for estimation of the total energy intake, the intake of 3 macronutrients (protein, fat, and carbohydrate), and the intake of 33 micronutrients. Two previous studies demonstrated that BDHQ-derived intake estimates are consistent with intake measured using semi-weighted 16-day dietary records [32,33]. The BDHQ has been used in cross-sectional [34] or cohort [35] studies. In the present study, we measured the dietary habits using the BDHQ at three time-points in a 2-year time period (May 2015 as baseline, May 2016, and May 2017).

### 2.3. Measurements of Other Items

All participants attended a health check-up early in the morning after fasting for at least 9 h at baseline and each subsequent year. VFA was measured using a visceral fat meter, the EW-FA90 (Panasonic Corporation, Osaka, Japan), which is an authorized medical device in Japan (No. 22500BZX00522000). This device produces results that highly correlate with those obtained using computed tomography (CT) [36], the gold standard for VFA measurement. As metabolic risk factors, the following clinical characteristics were measured: BMI (calculated from height and weight), WC, VFA, serum glucose, glycated hemoglobin (HbA1c), systolic blood pressure (SBP), diastolic blood pressure (DBP), serum TG, HDL-cholesterol, and low-density lipoprotein (LDL)-cholesterol. Blood samples were collected from peripheral veins. All laboratory tests were outsourced to LSI Medience Co. (Tokyo, Japan) according to the instructions of the vendors. Smoking habit (cigarettes/d), sleep time (h/d), and amount of exercise (mets/d) were determined from questionnaires. Amount of exercise was calculated using Mets conversion table by following questionnaire items; type of exercise, number of times per week, and hours per time [37].

### 2.4. Statistical Analysis

Characteristics of the study participants are reported as means ± standard deviation (SD). At baseline, the relationship between VFA and each value was tested for linearity using a multivariate model, including age, smoking habit, amount of exercise, BMI, and WC, which are well-known confounding factors of VFA and dietary habits. The amount of change in each nutrient and VFA was calculated from the slope of the three time-points obtained for each parameter. We hypothesized that a clearer association between micronutrients and VFA could be obtained by adjusting for related confounders. The associations between change in VFA and change in each macronutrient were assessed by a multiple regression analysis with change in VFA as the objective variable, and change in the macronutrient and covariates, such as age, sex, VFA, and amount of each macronutrient ingested at baseline; and change in the amount of exercise, BMI, WC, smoking habit, and energy ingested as explanatory variables. The associations between change in VFA and change in each micronutrient were assessed by multiple regression analysis with change in VFA as the objective variable, and change in each micronutrient and covariates, such as age, sex, VFA, and intake amount of each macronutrient (protein, fat, and carbohydrate) at baseline; and change in the amount of each macronutrient (protein, fat, and carbohydrate) ingested, amount of exercise, BMI, WC, smoking habit, and amount of energy ingested as explanatory variables. The associations between change in BMI and change in each micronutrient were assessed using multiple regression analysis with change in BMI as the objective variable, and change in each micronutrient and covariates, such as age, sex, BMI, and amount of each macronutrient (protein, fat, and carbohydrate) ingested at baseline; and change in the amount of each macronutrient (protein, fat, and carbohydrate) ingested, amount of exercise, VFA, WC, smoking habits, and amount of energy ingested as explanatory variables.

To investigate the association between the change in each macronutrient and the change in each micronutrient, the correlation coefficient and its P value were estimated by the Spearman correlation test. Statistical tests were two-sided, and values of *p* < 0.05 were considered statistically significant. All analyses were performed using the R software version 3.3.4.

### 2.5. Ethics Statement

This study was performed in accordance with the ethical standards of the Declaration of Helsinki and approved by the ethics committee at Hirosaki University Medical Ethics Committee (2014-377, 2016-028, and 2017-026). Written informed consent was obtained from all participants prior to the study. This study was registered at the University Hospital Medical Information Network (UMIN-CTR, https://www.umin.ac.jp) prior to the analyses (UMIN ID: UMIN000030351).

## 3. Results

### 3.1. Baseline Characteristics Based on the Cut-Off Point for Visceral Obesity

Baseline characteristics of the study sample (N  =  624, 58% female) are summarized in Table 1. The rate of overweight (defined as BMI  ≥ 25) was 29.2% in men and 16.2% in women. The rates were comparable to those of the 2010 national survey reported by the Japanese government (overweight and obesity rate in subjects 30–69 years of age: 33.5% in men and 20.5% in women). Mean VFA was 106.8 ± 42.7 cm^2^ in men and 66.4 ± 32.0 cm^2^ in women, and was higher in men and lower in women than the value defined as visceral obesity (≥100 cm^2^) by the Japan Society for the Study of Obesity [38]. The individuals were divided into two groups based on the cut-off point for VFA, the VFA+ group (VFA ≥ 100, 144 men and 45 women) or the VFA-group (VFA < 100, 116 men and 319 women; Table 1). As for metabolic risk factors, glucose, HbA1c, SBP, DBP, TG, and LDL cholesterol values were significantly higher in the VFA+ group, while the HDL cholesterol values were significantly lower regardless of sex (*p* < 0.001, <0.001, <0.001, 0.021, <0.001, <0.001, and 0.001 for men and <0.001, <0.001, <0.001, <0.001, <0.001, <0.001, and 0.001 for women, respectively). As for dietary habits, intake of energy or macronutrients was not significantly associated with VFA. Furthermore, in men, only retinol intake was significantly inversely associated with VFA (*p* = 0.048), and in women, only sodium intake was significantly positively associated with VFA (*p* = 0.033), but no other micronutrient examined was significantly associated with VFA, regardless of sex, in the baseline study.

### 3.2. Association Between Changes in VFA and Macronutrient Intake Over 2 Years

The association between changes in visceral fat and intake of macronutrients over 2 years is shown in Table 2 adjusted by Models 1 and 2. In Model 1, the adjustment was performed for the following factors: age, sex, VFA, and amount of each macronutrient (protein, fat, and carbohydrate) ingested at baseline, and amount of change in exercise, BMI, WC, and smoking habit. The change in VFA and in intake of energy, proteins, fats, or carbohydrates was not significantly associated (β = 0.002, *p* = 0.239, β = 0.043, *p* = 0.193, β= 0.057, *p* = 0.186, and β = 0.005, *p* = 0.632, respectively). In Model 2, change in energy intake was added to Model 1 as the confounding factor for the adjustment. Similar to the above-mentioned results, no significant relationship was detected between change in VFA and change in the intake amount of protein, fat, or carbohydrate (β = 0.033, *p* = 0.548, β = 0.042, *p* = 0.527, and β = −0.035, *p* = 0.108, respectively).

### 3.3. Association Between Macronutrients and Micronutrients

The associations between changes in macronutrients and micronutrients are shown in Table 3. Each macronutrient (protein, fat, and carbohydrate) significantly correlated with almost all of the micronutrients except for alcohol, suggesting that “macronutrients” and “micronutrients” could be mutual confounding factors. Therefore, we consider that the direct or indirect correlation with VFA cannot be estimated without adjusting for the other factor. That is, when examining the association of VFA with micronutrients or macronutrients, they must be adjusted for each other.

### 3.4. Macronutrients Associated with Change in VFA or BMI

The associations between changes in macronutrients and VFA or BMI are shown in Table 4. For VFA, in Model 1, the following factors were used for the adjustment: age, sex, VFA, and amount of each macronutrient (protein, fat, and carbohydrate) ingested at baseline, and the change in intake of energy, amount of exercise, BMI, WC, and smoking habit. Of the 33 micronutrients, changes in two nutrients, soluble dietary fiber and manganese, were significantly negatively associated with change in VFA (β = −2.31, *p* = 0.014 and β = −2.43, *p* = 0.018) over 2 years. In Model 2, the following factors were used for the adjustment: age, sex, VFA, and amount of each macronutrient (protein, fat, and carbohydrate) ingested at baseline, and the change in the amount of protein, fat, and carbohydrate ingested; amount of exercise; BMI; WC; and smoking habit. Of the 33 micronutrients, change in 7 micronutrients, soluble dietary fiber, manganese, potassium, magnesium, vitamin K, folic acid, and pantothenic acid, were significantly negatively associated with change in VFA (β = −2.59, *p* = 0.007, β = −2.19, *p* = 0.042, β = −0.004, *p* = 0.039, β = −0.06, *p* = 0.018, β = −0.011, *p* = 0.050, β = −0.018, *p* = 0.045 and β = −3.43, *p* = 0.001), and change in 1 micronutrient, monounsaturated fat, was significantly positively associated with change in VFA (β = 1.34, *p* = 0.050). However, vitamin K and monounsaturated fat had borderline p-values (*p* = 0.05). For BMI, in Model 3, the following factors were used for the adjustment: age, sex, BMI, and amount of each macronutrient (protein, fat, and carbohydrate) ingested at baseline, and change in the energy ingested, amount of exercise, VFA, WC, and smoking habit. Of the 33 micronutrients, none were significantly associated with a change in BMI. In Model 4, however, the following factors were used for the adjustment: age, sex, BMI, and amount of each macronutrient (protein, fat, and carbohydrate) ingested at baseline, and change in the amount of protein, fat, and carbohydrate ingested; amount of exercise; VFA; WC; smoking habit; and amount of energy ingested. Of the 33 micronutrients, changes in 4 nutrients: alpha-tocopherol, vitamin B6, vitamin B12, and n-3 polyunsaturated fat, were significantly negatively associated with change in VFA (β = −0.061, *p* = 0.023, β = −0.474, *p* = 0.015, β = −0.020, *p* = 0.015, and β = −0.115, *p* = 0.031). Taken together, 7 micronutrients: soluble dietary fiber, manganese, potassium, magnesium, vitamin K, folic acid, pantothenic acid, alpha-tocopherol, vitamin B6, vitamin B12, and n-3 polyunsaturated fat, which are abundant nutrients in a vegetable diet, had a significant inverse correlation to VFA. Only monounsaturated fat had a significant positive association with VFA. 

## 4. Discussion

We investigated the association of eating habits with change in VFA over 2 years. Baseline surveys indicated that VFA was significantly associated with other metabolic syndrome factors. These findings supported those of previous reports that VFA is a predictor of cardiovascular disease [4,5,6,7]. Regarding dietary habits, some nutrients were significantly associated with VFA in only men or in only women, but no nutrients were associated with VFA, regardless of sex, in our baseline analysis.

In our longitudinal analyses, we found no significant relationship between change in VFA and energy intake or amount of each macronutrient (protein, fat, and carbohydrate) ingested, even after 2 years, whether or not adjustments were made for changes in energy intake. Generally, habitual diets in humans are a mixture of “macronutrients” and “micronutrients”; therefore, we hypothesized that these two elements are mutual “confounding factors”, and thus determined their associations. The macronutrients were significantly highly correlated with all of the micronutrients except for alcohol (Table 3). Therefore, we adjusted for macronutrients when assessing the associations between micronutrients and VFA. In Model 1, which included changes in energy intake but not macronutrient intake, only two micronutrients: soluble dietary fiber and manganese, were significantly associated with VFA. In Model 2, which included changes in macronutrient intake but not energy intake, nine macronutrients were significantly associated with VFA (Table 4). Similar results were found when using BMI as an index of obesity. These findings suggest that different results may be obtained when investigating the association of micronutrients with VFA depending on whether adjustments are made for macronutrients. Although further studies are needed, this might be one reason that previous studies reported only a few macronutrients associated with VFA [21,22,23].

To gain insight into the importance of adjusting for macronutrients when investigating the association between micronutrients and VFA, the validity of our observation of each nutrient is discussed here and compared with the findings of previous reports. With respect to water-soluble dietary fiber, inconsistent conclusions were reported among previous studies [21,23,39,40]. A study of 85 individuals [23] reported no correlation; however, in a survey of 1114 individuals [21], those individuals who increased their intake of soluble dietary fiber had reduced visceral fat. In addition, an intervention trial reported that dietary fiber [39] and β-glucan [40], which is a type of soluble dietary fiber, are effective for reducing VFA, supporting our results. With regard to folic acid, no significant relationship was identified in an epidemiologic study [41], but an intervention study reported that folic acid intake was effective for reducing visceral fat [42]. The association between folic acid and VFA observed in the present study differed from the previous epidemiology study [41], which might be due to the differences in our analytical method. Magnesium reportedly contributes to insulin resistance and the risk of metabolic syndrome in association with visceral fat [43], possibly supporting our observations. With regard to manganese, intake of manganese was inversely correlated with abdominal obesity in a cross-sectional study [44]. Although an interventional test is required, our result is consistent with this report. With regard to vitamin K, which had a borderline *p*-value (*p* = 0.05), an interventional study reported that it is related to improvement of WC, and highly negatively correlated with VFA [45], supporting our results. No previous studies of pantothenic acid in relation to VFA have been reported. An intervention study of this association would be useful. In cross-sectional studies, no significant association between monounsaturated fatty acids, which had a borderline *p*-value (*p* = 0.05), and VFA are reported [21,46]. A longitudinal study found a significant inverse association [47], which supports the present results. Calcium and alcohol are reported to be significantly associated with VFA, but our study did not confirm this finding. Both a cross-sectional study [41] and a long-term study [22] reported that high or increased visceral fat is associated with a low calcium intake. While some interventional studies have evaluated the association between calcium and vitamin D3 intake with visceral fat [48,49], there are no human intervention studies investigating the association between calcium alone and visceral fat. Some cross-sectional studies report a significant relationship of alcohol with VFA [10,12,50,51], whereas others report no significant relationship [41,46]. An intervention study [52] found no significant relationship. Thus, the inconsistencies in the associations between micronutrients and VFA might be clarified by adjusting for macronutrients as confounding factors, even in epidemiologic studies.

On the other hand, the opposite associations are well known in animal meat. These associations are consistently reported for foodstuffs [53,54,55,56]. In our observations, however, the micronutrients associated with VFA and with BMI were completely different. This is the first study, to our knowledge, to evaluate nutrients associated with both VFA and with BMI. Thus, there might be unique associations of micronutrients with VFA in contrast to a simple obesity index like BMI. Of note, visceral fat is reported to be a higher risk factor for all-cause mortality, including cardiovascular disease, than BMI [4,5,6,7].

A limitation of the present study is that it was a 2-year longitudinal study, and the causality between food and obesity is unclear. As this study was performed in a limited country, region, and race, the reproducibility should be confirmed in a different country and/or race. VFA was measured using an impedance method rather than the gold standard method of CT, but the results of the impedance method correlate highly with those of CT [36]. The BDHQ estimates the nutritional components for only a small number of nutrients. Furthermore, a questionnaire regarding nutritional compounds during the preceding months is flawed with uncertainties.

The present study is a longitudinal study and reports on macronutrients associated with visceral fat using a comprehensive dataset of dietary micronutrients obtained with the BDHQ. Moreover, to further clarify the association of micronutrients with VFA, macronutrients were considered as confounders. Therefore, eight micronutrients were found to be significantly associated with VFA.

As visceral fat is a predictive factor of cardiovascular disease, it is important to analyze the factors involved in the development of visceral fat in healthy individuals. Thus, our observation in over 600 relatively healthy individuals is expected to be useful for proposing a healthy diet.

## 5. Conclusions

Seven micronutrients abundant nutrients in vegetable diets were found to potentially contribute to suppressive effects against the accumulation of visceral fat. Only one micronutrient, monounsaturated fat, was significantly positively associated with the accumulation of visceral fat. Analyses that adjust for macronutrients as confounders might be an alternative for evaluating the association of micronutrients with VFA.

## Figures and Tables

**Figure 1 nutrients-11-02698-f001:**
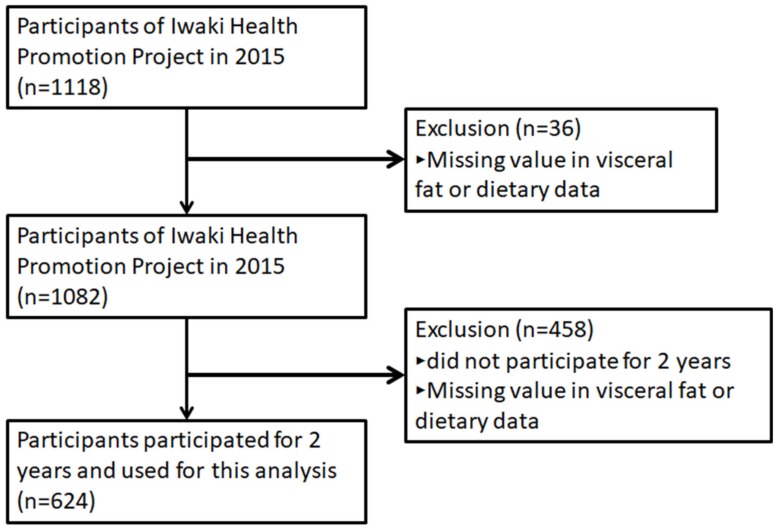
Study flow of the participants. A total of 624 participants completed measurements at the 3 time-points for 2 years (baseline in 2015, 2016, and 2017) among 1118 adults who participated in Iwaki Health Promotion Projects in 2015 (baseline) and were enrolled for the analysis.

**Table 1 nutrients-11-02698-t001:** Baseline participants’ characteristics based on the cut-off point of visceral obesity.

	MEN			WOMEN		
Characteristics	VFA < 100	100 ≤VFA	*p* ^a^	VFA < 100	100 ≤ VFA	*p* ^a^
Number of participants	116	144		319	45	
Age (yr)	52.1 ± 14.5	55.5 ± 13.8		54.5 ± 13.7	59.5 ± 11.6	-
Metabolic risk factors:						
Body mass index (kg/m2)	21.7 ± 2.0	25.2 ± 2.7	-	21.4 ± 2.7	27.1 ± 3.6	-
Waist circumference (cm)	76.9 ± 5.1	87.7 ± 6.6	-	71.8 ± 7.3	87.1 ± 7.7	-
Visceral fat mass (cm^2^)	6.9 ± 17.5	137.1 ± 28.8	-	57.9 ± 22.8	125.8 ± 24.6	-
Glucose (mmol/l)	4.5 ± 0.6	5.0 ± 1.1	<0.001 ***	4.4 ± 0.7	4.8 ± 0.7	<0.001 ***
HbA1c (%)	5.6 ± 0.4	5.9 ± 0.8	<0.001 ***	5.7 ± 0.5	6.0 ± 0.7	<0.001 ***
SBP (mmHg)	122.5 ± 18.1	128.2 ± 16.6	<0.001 ***	117.3 ± 16.4	130.2 ± 16.9	<0.001 ***
DBP (mmHg)	77.4 ± 12.6	79.2 ± 10.2	0.021 *	71.8 ± 10.3	76.3 ± 10.4	<0.001 ***
TG (mmol/l)	1.1 ± 0.7	1.7 ± 1.4	<0.001 ***	0.9 ± 0.4	1.2 ± 0.5	<0.001 ***
HDL-cholesterol (mmol/l)	1.7 ± 0.4	1.5 ± 0.5	<0.001 ***	1.9 ± 0.4	1.6 ± 0.3	<0.001 ***
LDL-cholesterol (mmol/l)	2.9 ± 0.7	3.1 ± 0.7	0.001 **	3.0 ± 0.8	3.2 ± 0.7	0.001 **
Lifestyle habits:						
Smoking habits (stick/d)	12.8 ± 17.8	11.5 ± 11.9	-	2.3 ± 5.3	2.1 ± 4.9	-
Sleep time (h/d)	7.0 ± 1.2	7.3 ± 1.2	-	6.7 ± 1.0	7.1 ± 0.9	-
Amount of exercise (Mets/d)	6.9 ± 17.5	8.4 ± 18.4	-	4.3 ± 10.7	6.0 ± 10.8	-
Dietary habits:						
Energy (kcal/d)	2189.1 ± 667.7	2107.3 ± 543.7	0.331	1655 ± 439	1752 ± 503	0.517
Macronutrient:						
Protein (g/d)	76.1 ± 29.6	74.5 ± 24.5	0.422	64.0 ± 22.1	67.8 ± 25.8	0.489
Fat (g/d)	55.2 ± 22.0	54.5 ± 18.3	0.645	49.3 ± 17.3	50.0 ± 20.6	0.473
Carbohydrate (g/d)	303.1 ± 101.9	281.8 ± 82.4	0.115	225.8 ± 63.0	242.8 ± 64.9	0.130
Micronutrient:						
Ash (g/d)	19.6 ± 6.7	19.6 ± 5.7	0.647	16.5 ± 5.0	17.8 ± 6.0	0.184
Sodium (mg/d)	4864 ± 1672	4824 ± 1331	0.421	3845 ± 1145	4332 ± 1445	0.033 *
Potassium (mg/d)	2394 ± 976	2416 ± 929	0.860	2224 ± 823	2208 ± 846	0.977
Calcium (mg/d)	483 ± 209	519 ± 236	0.539	480 ± 196	490 ± 210	0.666
Magnesium (mg/d)	269 ± 99	267 ± 89	0.692	232 ± 78	237 ± 85	0.633
Phosphorus (mg/d)	1109 ± 420	1105 ± 373	0.614	957 ± 331	1005 ± 387	0.481
Iron (mg/d)	8.0 ± 3.6	7.8 ± 2.8	0.495	7.1 ± 2.6	7.2 ± 2.9	0.839
Zinc (mg/d)	9.0 ± 3.2	8.8 ± 2.5	0.601	7.5 ± 2.3	7.9 ± 2.6	0.522
Copper (mg/d)	1.3 ± 0.5	1.3 ± 0.4	0.503	1.1 ± 0.3	1.1 ± 0.3	0.410
Manganese (mg/d)	3.3 ± 1.2	3.2 ± 1.1	0.408	2.7 ± 0.9	2.9 ± 1.0	0.367
Vitamin A (retinol equivalent) (µg/d)	732 ± 861	626 ± 333	0.076	584 ± 321	559 ± 343	0.406
Retinol (µg/d)	503 ± 779	397 ± 263	0.048^*^	321 ± 230	345 ± 272	0.804
β-carotene equivalent (µg/d)	2706 ± 2060	2710 ± 2012	0.239	3122 ± 2111	2536 ± 1811	0.978
Vitamin D (µg/d)	15.1 ± 10.7	14.3 ± 9.8	0.155	13.4 ± 9.2	14.3 ± 11.9	0.310
α-tocopherol (mg/d)	7.1 ± 3.3	6.9 ± 2.5	0.676	6.5 ± 2.5	6.5 ± 2.7	0.544
Vitamin K (µg/d)	320.3 ± 172.0	322.0 ± 167.1	0.633	303 ± 161	282 ± 143	0.484
Thiamin (mg/d)	0.7 ± 0.3	0.7 ± 0.3	0.798	0.7 ± 0.2	0.7 ± 0.3	0.797
Riboflavin (mg/d)	1.3 ± 0.6	1.3 ± 0.5	0.694	1.2 ± 0.4	1.2 ± 0.4	0.839
Niacin (mg/d)	19.2 ± 8.2	18.2 ± 7.7	0.219	15.6 ± 6.2	15.6 ± 6.8	0.812
Vitamin B_6_ (mg/d)	1.3 ± 0.6	1.3 ± 0.5	0.582	1.1 ± 0.4	1.1 ± 0.5	0.805
Vitamin B_12_ (µg/d)	11.5 ± 8.3	10.4 ± 6.1	0.091	9.1 ± 5.5	9.7 ± 6.4	0.324
Folate (µg/d)	317.1 ± 158.7	306.3 ± 130.2	0.616	282 ± 126	274 ± 117	0.659
Pantothenic acid (mg/d)	6.8 ± 2.6	6.8 ± 2.2	0.848	5.9 ± 2.0	6.1 ± 2.1	0.658
Vitamin C (mg/d)	83.6 ± 52.2	86.0 ± 53.2	0.584	85.9 ± 47.6	87.9 ± 49.0	0.804
Saturated fat (g/d)	14.4 ± 6.1	14.3 ± 5.6	0.584	13.1 ± 5.2	13.4 ± 6.1	0.822
Monounsaturated fat (g/d)	19.6 ± 8.0	19.3 ± 6.8	0.706	17.3 ± 6.2	17.5 ± 7.7	0.326
Polyunsaturated fat (g/d)	14.0 ± 5.7	14.0 ± 4.4	0.840	12.4 ± 4.2	12.4 ± 4.6	0.296
n-3 polyunsaturated fat (g/d)	2.9 ± 1.4	2.8 ± 1.1	0.315	2.6 ± 1.1	2.6 ± 1.2	0.755
n-6 polyunsaturated fat (g/d)	11.1 ± 4.4	11.1 ± 3.5	0.941	9.8 ± 3.3	9.8 ± 3.6	0.218
Cholesterol (mg/d)	382.7 ± 189.1	381.5 ± 156.4	0.790	331 ± 148	342 ± 181	0.766
Total dietary fiber (g/d)	11.8 ± 5.4	11.7 ± 4.7	0.977	10.8 ± 4.2	11.0 ± 4.2	0.705
Soluble dietary fiber (g/d)	2.9 ± 1.5	2.9 ± 1.3	0.950	2.7 ± 1.1	2.7 ± 1.1	0.942
Insoluble dietary fiber (g/d)	8.5 ± 3.7	8.4 ± 3.2	0.990	7.8 ± 2.9	7.9 ± 2.9	0.682
Alcohol (g/d)	19.5 ± 21.2	22.1 ± 22.6	0.236	4.2 ± 10.6	5.0 ± 11.6	0.517
Daidzein (mg/d)	15.8 ± 10.3	16.7 ± 10.0	0.714	15.3 ± 9.2	15.8 ± 8.4	0.700
Genistein (mg/d)	26.7 ± 17.3	28.3 ± 17.0	0.720	25.9 ± 15.5	26.8 ± 14.2	0.684

*p* < 0.05, *p* < 0.01 and *p* < 0.001 are indicated by “*”, “**”, and “***”, respectively. Data are mean ± SD. ^a^
*p* value was adjusted by age, smoking habit, amount of exercise, BMI, and WC. Abbreviations: VFA, visceral fat area. BMI, body mass index. WC, waist circumference. SBP, systolic blood pressure. DBP, diastolic blood pressure, TG, triglyceride. HDL-cholesterol, high-density lipoprotein- cholesterol. LDL- cholesterol, low-density lipoprotein- cholesterol.

**Table 2 nutrients-11-02698-t002:** Association between VFA and macronutrient change over 2 years.

	Change in VFA			Adjusted by Model 1 ^a^			Adjusted by Model 2 ^b^		
Characteristics	Q1	Q2	Q3	β (s.e.)		*p*	β (s.e.)		*p*
Number of participants	203(79/124)	213(67/146)	208(114/94)						
Baseline age (y)	55.0 ± 13.6	54.7 ± 14.6	54.2 ± 13.3						
Visceral fat mass (Δcm^2^)	−9.1 ± 5.2	0.6 ± 2.2	10.9 ± 6.7						
Energy (Δkcal/d)	−1.5 ± 226.4	15.0 ± 199.7	21.2 ± 224.3	0.002	(0.002)	0.239	-	-	-
Macronutrient:									
Protein (Δg/d)	0.72 ± 10.80	2.29 ± 10.26	0.99 ± 11.17	0.043	(0.033)	0.193	0.033	(0.055)	0.548
Fat (Δg/d)	0.62 ± 8.78	0.97 ± 7.78	0.82 ± 8.56	0.057	(0.043)	0.186	0.042	(0.066)	0.527
Carbohydrate (Δg/d)	−2.83 ± 31.90	−1.10 ± 29.38	0.99 ± 34.22	0.005	(0.011)	0.632	−0.035	(0.022)	0.108

*p* < 0.05, *p* < 0.01 and *p* < 0.001 are indicated by “*”, “**” and “***”, respectively. Data are mean ± SD. β was regression coefficient. ^a^ Model 1 was adjusted by age, sex, VFA, and amount of each macronutrient (protein, fat, and carbohydrate) ingested at baseline, and the change in the amount of exercise, BMI, WC, and smoking habit. ^b^ Model 2 was adjusted by age, gender, VFA, and amount of each macronutrient (protein, fat, and carbohydrate) ingested at baseline, and the change in the amount of exercise, BMI, WC, smoking habit, and energy intake.

**Table 3 nutrients-11-02698-t003:** Association between Macronutrient and Micronutrient change over 2 years.

	Macronutrient		
	Protein	Fat	Carbohydrate
Characteristics	Correlation Coefficient (r)^a^		
Micronutrient:			
Ash (Δg/d)	0.88 ***	0.71 ***	0.54 ***
Sodium (Δmg/d)	0.78 ***	0.61 ***	0.50 ***
Potassium (Δmg/d)	0.79 ***	0.70 ***	0.49 ***
Calcium (Δmg/d)	0.69 ***	0.60 ***	0.39 ***
Magnesium (Δmg/d)	0.85 ***	0.70 ***	0.56 ***
Phosphorus (Δmg/d)	0.96 ***	0.81 ***	0.53 ***
Iron (Δmg/d)	0.84 ***	0.72 ***	0.48 ***
Zinc (Δmg/d)	0.92 ***	0.79 ***	0.63 ***
Copper (Δmg/d)	0.80 ***	0.61 ***	0.72 ***
Manganese (Δmg/d)	0.43 ***	0.32 ***	0.60 ***
Vitamin A (retinol equivalent) (Δµg/d)	0.45 ***	0.45 ***	0.21 ***
Retinol (Δµg/d)	0.47 ***	0.46 ***	0.19 ***
β-carotene equivalente (Δµg/d)	0.37 ***	0.32 ***	0.21 ***
Vitamin D (Δµg/d)	0.70 ***	0.51 ***	0.21 ***
α-tocopherol (Δmg/d))	0.79 ***	0.84 ***	0.41 ***
Vitamin K (Δµg/d)	0.50 ***	0.47 ***	0.25 ***
Thiamin (Δmg/d)	0.85 ***	0.80 ***	0.52 ***
Riboflavin (Δmg/d)	0.82 ***	0.77 ***	0.41 ***
Niacin (Δmg/d)	0.85 ***	0.73 ***	0.36 ***
Vitamin B6 (Δmg/d)	0.87 ***	0.76 ***	0.46 ***
Vitamin B12 (Δmg/d)	0.71 ***	0.53 ***	0.19 ***
Folate (Δµg/d)	0.58 ***	0.52 ***	0.37 ***
Pantothenic acid (Δmg/d)	0.90 ***	0.80 ***	0.58 ***
Vitamin C (Δmg/d)	0.44 ***	0.41 ***	0.35 ***
Saturated fat (Δg/d)	0.74 ***	0.91 ***	0.42 ***
Monounsaturated fat (Δg/d)	0.79 ***	0.98 ***	0.38 ***
Polyunsaturated fat (Δg/d)	0.77 ***	0.90 ***	0.38 ***
n-3 polyunsaturated fat (Δg/d)	0.78 ***	0.81 ***	0.30 ***
n-6 polyunsaturated fat (Δg/d)	0.72 ***	0.87 ***	0.39 ***
Cholesterol (Δmg/d)	0.80 ***	0.72 ***	0.32 ***
Total dietary fiber (Δg/d)	0.59 ***	0.49 ***	0.56 ***
Soluble dietary fiber (Δg/d)	0.54 ***	0.46 ***	0.47 ***
Insoluble dietary fiber (Δg/d)	0.59 ***	0.49 ***	0.59 ***
Alcohol (Δg/d)	0.02	0.07	0.06
Daidzein (Δmg/d)	0.42 ***	0.35 ***	0.21 ***
Genistein (Δmg/d)	0.42 ***	0.35 ***	0.21 ***

*p* < 0.001 are indicated by “***”, respectively. Data are shown as mean ± SD. ^a^ Correlation coefficient and *p* value were analyzed by Spearman’s rank partial correlation analysis.

**Table 4 nutrients-11-02698-t004:** Association between VFA and micronutrient change over 2 years.

	Change in VFA						Change in BMI					
Characteristics	Adjusted by Model 1			Adjusted by Model 2			Adjusted by Model 3			Adjusted by Model 4		
Micronutrient:	β (s.e.)		*p*	β (s.e.)		*p*	β (s.e.)		*p*	β (s.e.)		*p*
Ash (Δg/d)	−0.03	(0.24)	0.904	−0.32	(0.36)	0.369	−0.001	(0.011)	0.956	−0.005	(0.016)	0.727
Sodium (Δmg/d)	0.00	(0.00)	0.694	0.00	(0.00)	0.949	0.000	(0.000)	0.957	0.000	(0.000)	0.965
Potassium (Δmg/d)	−0.00	(0.00)	0.250	−0.01	(0.00)	0.039 *	0.000	(0.000)	0.682	0.000	(0.000)	0.455
Calcium (Δmg/d)	0.00	(0.00)	0.372	−0.01	(0.01)	0.134	0.000	(0.000)	0.524	0.000	(0.000)	0.605
Magnesium (Δmg/d)	−0.02	(0.02)	0.209	−0.06	(0.02)	0.018 *	0.000	(0.001)	0.565	−0.001	(0.001)	0.272
Phosphorus (Δmg/d)	0.00	(0.00)	0.951	−0.01	(0.01)	0.111	0.000	(0.000)	0.797	0.000	(0.000)	0.724
Iron (Δmg/d)	−0.34	(0.44)	0.445	−1.06	(0.61)	0.084	−0.004	(0.019)	0.673	−0.019	(0.027)	0.471
Zinc (Δmg/d)	0.26	(0.65)	0.684	−0.20	(1.00)	0.844	0.082	(0.044)	0.062	0.082	(0.044)	0.062
Copper (Δmg/d)	−7.34	(4.45)	0.100	−10.4	(5.39)	0.055	−0.089	(0.197)	0.650	−0.133	(0.239)	0.578
Manganese (Δmg/d)	−2.43	(1.02)	0.018	−2.19	(1.07)	0.042 *	0.000	(0.001)	0.565	−0.001	(0.001)	0.272
Vitamin A (retinol equivalent) (Δµg/d)	0.00	(0.00)	0.582	0.00	(0.00)	0.436	0.000	(0.000)	0.980	0.000	(0.000)	0.938
Retinol (Δµg/d)	0.00	(0.00)	0.618	0.00	(0.00)	0.501	0.000	(0.000)	0.940	0.000	(0.000)	0.898
β-carotene equivalente (Δµg/d)	0.00	(0.00)	0.885	0.00	(0.00)	0.726	0.000	(0.000)	0.992	0.000	(0.000)	0.944
Vitamin D (Δµg/d)	0.07	(0.08)	0.423	0.09	(0.12)	0.452	−0.003	(0.004)	0.375	−0.010	(0.005)	0.063
β-tocopherol (Δmg/d)	0.44	(0.43)	0.307	0.34	(0.62)	0.578	−0.023	(0.019)	0.219	−0.061	(0.027)	0.023 *
Vitamin K (Δµg/d)	−0.01	(0.01)	0.131	−0.01	(5.85)	0.050 *	0.000	(0.000)	0.360	0.000	(0.000)	0.395
Thiamin (Δmg/d)	−2.58	(5.27)	0.625	−11.9	(7.48)	0.111	0.112	(0.232)	0.629	0.085	(0.330)	0.797
Riboflavin (Δmg/d)	0.12	(0.12)	0.339	0.19	(0.18)	0.294	0.134	(0.112)	0.230	0.184	(0.151)	0.224
Niacin (Δmg/d)	0.12	(0.15)	0.417	0.18	(0.27)	0.522	−0.001	(0.007)	0.930	−0.010	(0.012)	0.426
Vitamin B_6_ (Δmg/d)	0.54	(2.61)	0.837	−2.14	(4.48)	0.632	−0.117	(0.114)	0.307	−0.474	(0.195)	0.015 *
Vitamin B_12_ (Δmg/d)	0.12	(0.12)	0.339	0.19	(0.18)	0.294	−0.006	(0.005)	0.260	−0.020	(0.008)	0.015 *
Folate (Δµg/d)	−0.01	(0.01)	0.116	−0.02	(0.01)	0.045 *	0.000	(0.000)	0.783	0.000	(0.000)	0.655
Pantothenic acid (Δmg/d)	−1.16	(0.70)	0.099	−3.43	(1.02)	0.001 **	0.049	(0.031)	0.115	0.089	(0.046)	0.051
Vitamin C (Δmg/d)	−0.02	(0.02)	0.448	−0.02	(0.02)	0.341	−0.002	(0.001)	0.068	−0.002	(0.001)	0.057
Saturated fat (Δg/d)	−0.06	(0.21)	0.781	−0.48	(0.35)	0.170	0.013	(0.009)	0.165	0.025	(0.015)	0.105
Monounsaturated fat (Δg/d)	0.16	(0.17)	0.325	1.34	(0.68)	0.050 *	0.006	(0.007)	0.431	0.016	(0.030)	0.605
Polyunsaturated fat (Δg/d)	0.16	(0.24)	0.497	0.03	(0.42)	0.938	−0.001	(0.011)	0.900	−0.019	(0.018)	0.287
n-3 polyunsaturated fat (Δg/d)	0.71	(0.82)	0.389	0.57	(1.22)	0.640	−0.034	(0.036)	0.347	−0.115	(0.053)	0.031
n-6 polyunsaturated fat (Δg/d)	0.14	(0.30)	0.642	−0.10	(0.48)	0.836	0.004	(0.013)	0.770	−0.004	(0.021)	0.864
Cholesterol (Δmg/d)	0.00	(0.01)	0.772	0.00	(0.01)	0.850	0.000	(0.000)	0.318	0.000	(0.000)	0.438
Total dietary fiber (Δg/d)	−0.64	(0.28)	0.022	−0.73	(0.29)	0.013 *	−0.012	(0.012)	0.320	−0.014	(0.013)	0.298
Soluble dietary fiber (Δg/d)	−2.31	(0.94)	0.014	−2.59	(0.97)	0.007 **	−0.046	(0.041)	0.264	−0.050	(0.043)	0.245
Insoluble dietary fiber (Δg/d)	−0.74	(0.41)	0.070	−0.82	(0.43)	0.055	−0.017	(0.018)	0.333	−0.019	(0.019)	0.313
Alcohol (Δg/d)	−0.03	(0.05)	0.584	−0.01	(0.05)	0.908	0.001	(0.002)	0.541	0.001	(0.002)	0.497
Daidzein (Δmg/d)	−0.11	(0.09)	0.205	−0.14	(0.09)	0.126	0.003	(0.004)	0.405	0.003	(0.004)	0.428
Genistein (Δmg/d)	−0.07	(0.05)	0.213	−0.08	(0.05)	0.132	0.002	(0.002)	0.414	0.002	(0.002)	0.438

*p* < 0.05 and < 0.01 are indicated by “*” and “**” respectively. β was regression coefficient. ^a^ Model 1 was adjusted by age, gender, VFA, and intake amount of each macronutrient (protein, fat, and carbohydrate) at baseline, and change in energy intake, amount of exercise, BMI, WC, and smoking habit. ^b^ Model 2 was adjusted by age, sex, VFA, and intake amount of each macronutrient (protein, fat, and carbohydrate) at baseline, and change in intake of protein, fat, and carbohydrate; amount of exercise; BMI; WC; smoking habit; and amount of energy. ^c^ Model 3 was adjusted by age, gender, BMI, and intake amount of each macronutrient (protein, fat, and carbohydrate) at baseline, and change in energy intake, amount of exercise, VFA, WC, and smoking habit. ^d^ Model 4 was adjusted by age, gender, BMI, and amount of each macronutrient at baseline, and change in intake of protein, fat, and carbohydrate; amount of exercise; VFA; WC; smoking habit; and amount of energy.

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
