# Peer review of "Association between Nutrients and Visceral Fat in Healthy Japanese Adults: A 2-Year Longitudinal Study Brief Title: Micronutrients Associated with Visceral Fat Accumulation"

_nutrients, 2019, doi:10.3390/nu11112698_

Round 1

Reviewer 1 Report

line 40: suboptimal diet is responsible for more deaths than any other risk globally. Please specify in this sentence that this statement is focused on a global view. Individually smoking may be more dangerous than unhealthy eating habits.

line 52: please correct individual

line 56: please clarify the statement: "relatively healthy Japanese subjects". Were there exclusion criteria, what makes a person not to be relatively healthy? In the title you state, "healthy Japanese adults" were included.

line 98: which questionnaire was used to assess physical activity?

line 198. 7 micronutrients have a "significant positive association to VFA". However, ß is negative, the correlation is inverse. This may be confusing, I would prefer to change the sentence to "7 micronutrients have a inverse correlation to VFA".

line 261: the BDHQ estimates the nutritional components for only a small number of nutrients. Furthermore, a questionnaire regarding nutritional compounds during the preceding months are flawed with uncertanties. Please mention this as a weakness

Author Response

Responses to the comments made by Reviewer 1

Thank you very much for your suggestions and comments, which have helped us to greatly improve the quality of our paper. We seriously considered each of the comments and revised the manuscript according to the suggestions, as described below.

line 40: suboptimal diet is responsible for more deaths than any other risk globally. Please specify in this sentence that this statement is focused on a global view. Individually smoking may be more dangerous than unhealthy eating habits.

Based on your suggestion, we changed the Abstract as follows:

“Dietary habits are crucial in the development of overweight and obesity [2], and suboptimal diet can potentially be a major contributor to mortality in all countries worldwide [3].”

line 52: please correct individual

As suggested, we changed the word “individual” as “individuals”.

line 56: please clarify the statement: "relatively healthy Japanese subjects". Were there exclusion criteria, what makes a person not to be relatively healthy? In the title you state, "healthy Japanese adults" were included.

As suggested, more clear description is now added based on our study protocol.

The Iwaki Health Promotion Project is a health check-up that takes more than 5 hours including physical fitness measurements. Therefore, inclusion criterion has been defined as follows; individuals who have health condition that enables to participate in this health check-up for more than 5 hours. The following statement has been added in the Method section.

“Inclusion criterion was individuals who have health condition ability to participate in this health check-up for more than 5 hours.”

Additionally, together with the other reviewer’s suggestion, we have revised Title as follows:

“Association between nutrients and visceral fat in healthy Japanese adults: a 2-year longitudinal study”

line 98: which questionnaire was used to assess physical activity?

The questionnaire lists the type of exercise, number of times per week, and hours of an exercise. From this information, the amount of exercise is calculated by referring to the Mets conversion table for Japanese.

As suggested, we have added the following sentence to the Method section:

“Amount of exercise was calculated using Mets conversion table by following questionnaire; type of exercise, number of times per week, and hours per time [37].”

[37] Med Sci Sports Exerc. 2011 Aug;43(8):1575-81. doi: 10.1249/MSS.0b013e31821ece12.

line 198. 7 micronutrients have a "significant positive association to VFA". However, ß is negative, the correlation is inverse. This may be confusing, I would prefer to change the sentence to "7 micronutrients have a inverse correlation to VFA".

As suggested, we have changed the sentence in the Results section as follows:

“…had a significant inverse correlation to VFA.”

line 261: the BDHQ estimates the nutritional components for only a small number of nutrients. Furthermore, a questionnaire regarding nutritional compounds during the preceding months are flawed with uncertainties. Please mention this as a weakness.

As suggested, we have added the following discussion in page XX:

“The BDHQ estimates the nutritional components for only a small number of nutrients. Furthermore, a questionnaire regarding nutritional compounds during the preceding months are flawed with uncertainties.”

Reviewer 2 Report

Longitudinal study of a sample of healthy Japanese adults that aims to relate the consumption of certain nutrients with changes in visceral fat. The manuscript presents interesting novel findings for readers.

The following changes are suggested:
Shortening the length of the article title is proposed: Association between nutrients and visceral fat in healthy Japanese adults: a 2-year longitudinal study

Because it has been established as statistical significance the values of P <0.05 nutrients (vitamin K, Munsaturated fat and that have borderline values (P = 0.05) in contrast tests should be considered differentially in the results and discussion.

Author Response

Responses to the comments made by Reviewer 2

Thank you very much for your suggestions and comments, which we feel have helped us to greatly improve the quality of our paper. We seriously considered your comments and revised the manuscript according to the suggestions, as described below.

Longitudinal study of a sample of healthy Japanese adults that aims to relate the consumption of certain nutrients with changes in visceral fat. The manuscript presents interesting novel findings for readers.

The following changes are suggested:

Shortening the length of the article title is proposed: Association between nutrients and visceral fat in healthy Japanese adults: a 2-year longitudinal study

As suggested, we have changed the title as follows:

“Association between nutrients and visceral fat in healthy Japanese adults: a 2-year longitudinal study”

Because it has been established as statistical significance the values of P <0.05 nutrients (vitamin K, Munsaturated fat and that have borderline values (P = 0.05) in contrast tests should be considered differentially in the results and discussion.

As suggested, we added the following sentence in Results:

“However, vitamin K and monounsaturated fat had borderline p-values (P = 0.05).”

As suggested, we changed the Discussion as follows:

“With regard to vitamin K, which had a borderline p-value (P = 0.05), ...”

“… no significant association between monounsaturated fatty acids, which had a borderline p-value (P = 0.05), and VFA are reported ...”
